# Water acting as a catalyst for electron-driven molecular break-up of tetrahydrofuran

Enliang Wang [1], Xueguang Ren [1,2 ✉], WoonYong Baek[3], Hans Rabus [3], Thomas Pfeifer [1] & Alexander Dorn[1]

Low-energy electron-induced reactions in hydrated molecular complexes are important in various fields ranging from the Earth's environment to radiobiological processes including radiation therapy. Nevertheless, our understanding of the reaction mechanisms in particular in the condensed phase and the role of water in aqueous environments is incomplete. Here we use small hydrogen-bonded pure and mixed dimers of the heterocyclic molecule tetrahydrofuran (THF) and water as models for biochemically relevant systems. For electron-impact-induced ionization of these dimers, a molecular ring-break mechanism is observed, which is absent for the THF monomer. Employing coincident fragment ion mass and electron momentum spectroscopy, and theoretical calculations, we find that ionization of the outermost THF orbital initiates significant rearrangement of the dimer structure increasing the internal energy and leading to THF ring-break. These results demonstrate that the local environment in form of hydrogen-bonded molecules can considerably affect the stability of molecular covalent bonds.

[1] Max-Planck-Institut für Kernphysik, 69117 Heidelberg, Germany. [2] School of Science, Xi'an Jiaotong University, 710049 Xi'an, China. [3] Physikalisch-Technische Bundesanstalt, 38116 Braunschweig, Germany. ✉email: renxueguang@xjtu.edu.cn

I n recent years, there has been intense research on the microscopic interactions of energetic radiation with organic and biological relevant molecules. An important motivation is the application of radiation in medical treatment like radiation therapy and the desire to understand the underlying mechanisms and possibly to improve its effectiveness[1]. The primary X-rays or swift charged particles, while penetrating biological tissue, produce large numbers of secondary electrons that in turn cause cellular damage either directly via ionization or indirectly by producing radicals such as OH in the aqueous environment[2–5]. To get insight into the reaction mechanisms, on one hand, gas-phase experiments were conducted on building blocks of macro-molecules such as DNA or proteins, where bulk effects do not mask the intrinsic molecular properties. On the other hand, one has to recognize the influence of the natural environment on biomolecules, in particular through hydrogen bonding, which can modify their structure and functionality. Examples are the hydrogen bonds between DNA base pairs linking both strands and the structural water molecules, which are hydrogen-bonded to the DNA with roughly 22 hydration sites per base pair[6].

Consequently, a number of studies on pure and nano-hydrated biomolecular clusters have been performed using different mass spectrometric techniques for both cations and anions[7–16]. The effect of solvation on the fragmentation of biomolecules after collisions with various projectiles was studied by measuring the yields of different fragment ions. A general observation was that although monomer ionization results in a large number of frag-mentation channels for larger clusters, essentially only cations with integer number of intact molecules were found in the mass spectra. It was concluded that the environment has an overall "protective" effect on the systems and that the cluster environ-ment acts as a buffer that rapidly redistributes excess energy, leading to suppression of molecular dissociation in clusters. On the other hand, for the smallest clusters, i.e., for dimers and tri-mers, some new molecular fragment species can be identified in the mass spectra, which are not consistent with the aforemen-tioned protection effect[9–12]. The authors did not discuss or clarify the formation mechanisms of these species. Therefore, in the present study we go beyond pure fragment mass measurements and identify the initial ionized states from which the fragmen-tation process starts. This information accompanied by high-level quantum-chemistry calculations gives insight into the molecular geometry evolution and intermediate transition states (TS), which must be overcome.

We investigate the electron-collision-induced ionization and dissociation processes in clusters consisting of water and tetra-hydrofuran (THF). Here, the THF ($C_4H_8O$) molecule is considered as a molecular analog of the deoxyribose sugar-ring in the DNA backbone (see Fig. 1a). It provides a simple model to probe possible mechanisms of electron-induced deoxyribose decomposition[17–19]. Water ($H_2O$) is the predominant medium in which biological chemistry takes place[20]. An accurate description of the energetic and structural aspects of the interaction of water with biomolecules is essential for a better understanding of their functions in biolo-gical processes[21–28]. The present study aims to understand how the reaction properties of the isolated THF molecule are affected when a water or second THF molecule is attached to it via hydrogen bonding, to mimic a chemical environment. We elucidate the electron-driven fragmentation dynamics of hydrated and pure THF clusters, i.e., $H_2O \cdot THF$ and $THF \cdot THF$ dimers (Fig. 1b, c) in com-parison with the isolated THF molecule.

Experiments were carried out using a multi-particle (electrons plus ions) imaging spectrometer with a supersonic gas jet target and a pulsed low-energy electron beam[29,30]. The projectile energy of 65 eV was chosen close to the mean energy of secondary electrons produced by high-energy primary radiation in a condensed medium such as water[4]. We find that the removal of an electron from the highest occupied molecular orbital (HOMO), which leads to a stable parent ion for the THF monomer, for the dimer initiates a THF ring-break reaction. Our ab-initio calcula-tions show that THF ring-break after HOMO ionization requires geometrical changes via several TSs, including those requiring structural rearrangement like ring-opening and proton transfer (PT). The highest barrier to overcome is a C–C bond-break for the ring-opening. For the THF molecule embedded in a $THF \cdot THF$ or $H_2O \cdot THF$ dimer, the energy of the respective TS is reduced in comparison with the isolated THF molecule. This reduced barrier can activate ring-break for the dimers, while this channel is closed in the isolated THF molecule. In addition, PT takes place during the molecular relaxation, which releases some amount of internal energy to the system. As a consequence, the cluster cation finally dissociates, i.e., $H_2O \cdot THF^+ \rightarrow H_2O \cdot C_2H_4O^+ + C_2H_4$, (see in Fig. 1d). These observations reveal a so-far unnoticed role of the water environment in enhancing the ring-break of the THF molecule after ionization of the HOMO. It can be inferred that noncovalent hydrogen bonding can considerably weaken the covalent bonds in a neighboring molecule. This can be important for a better understanding of the reaction mechanisms concerning ionizing radiation in biological matter[3,5].

## Results

**Sample composition and characterization of reaction products**. In our experiments, two kinds of gas jet targets were employed, a pure THF jet containing about 10–15% dimers $THF \cdot THF$ and a mixed THF water jet with about 4% THF dimers and 4% $H_2O \cdot THF$ dimers. The abundance of larger clusters goes down by a factor of roughly 5 for each additional molecule (these numbers are discussed in Methods). Therefore, most of the ionizing colli-sions concern monomers and the identification of dimer ionization processes is according to the characteristic mass of the ionic fragment species. For each ionizing collision, the ion and the two outgoing electrons are detected in coincidence. During offline analysis, the mass-over-charge ratios, the momentum vectors and the kinetic energies for all three charged particles are determined (see Methods). In case there is not more than one neutral frag-ment, its momentum can also be reconstructed from the measured momenta and momentum conservation, and the measurement is kinematically complete. We deduce the correlation of the ionic fragment species with the ionized electron's binding energy (BE) from which the ionized orbital is identified. Here, the BE $E_b$ is determined as the initial projectile energy $E_0$ minus the sum energy of the two final state electrons $E_1 + E_2$, i.e., $E_b = E_0 - (E_1 + E_2)$. $E_b$ constitutes the vertical transition energy between the electronic ground state and an ionized state of the molecule.

**Fragment mass spectra for monomers and clusters**. The mea-sured mass spectra of pure and hydrated THF clusters are pre-sented in Fig. 2 in the range from 10 to 150 u (atomic mass units). The spectra are normalized at mass 72 u corresponding to the intact $C_4H_8O^+$ cation. Hydrogen abstraction from THF mono-mers gives rise to $C_4H_7O^+$. Ring-break reactions in THF mono-mers yield the ions $C_2H_n^+$, $C_3H_n^+$, and $C_2H_nO^+$ assigned in Fig. 2. The fragmentation of the THF ion was studied before by us and other groups[12,31]. Briefly, ionization of the HOMO gives rise to a stable parent ion while hydrogen abstraction occurs for HOMO-1 ionization which is 1.5 eV above the ionic ground state. The behavior of the ring-break channels complies with an unimolecular statistical decay: electronically excited states pro-duced by ionization of inner orbitals quickly evolve to the ionic ground state (internal conversion) giving rise to vibrational exci-tation. The excess energy of 2.5 eV is sufficient for ring-opening

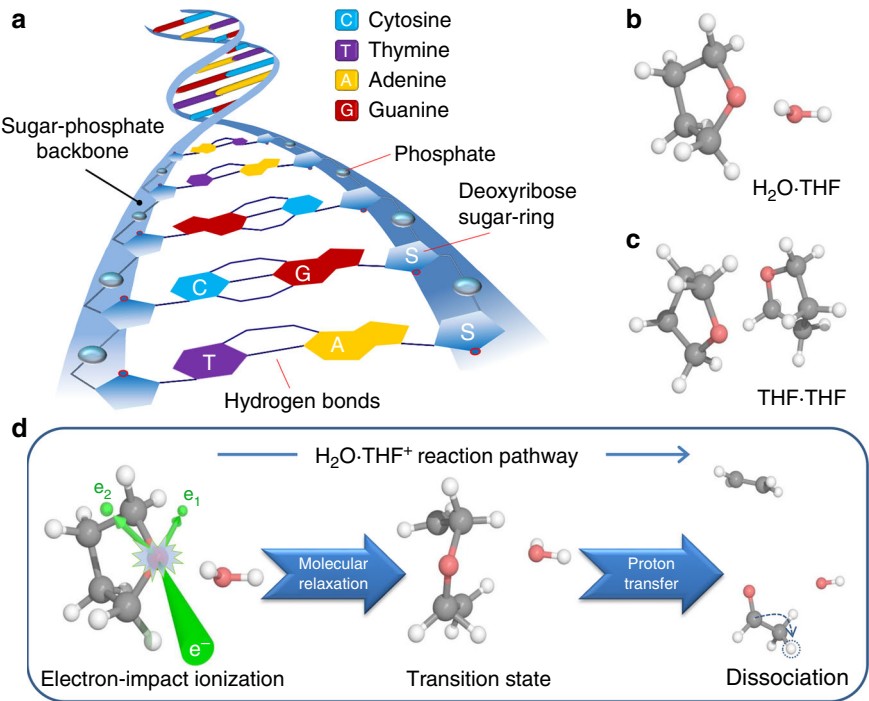

**Fig. 1 Chemical structure of the studied systems and schematic of the ring-break process. a** A section of DNA containing the four bases and the sugar-phosphate backbone. **b, c** Chemical structure of $H_2O\cdot THF$ (**b**) and $THF\cdot THF$ (**c**) dimers for the hydrated and pure THF model systems. **d** Schematic of the electron-induced ionization and ring-break process in the $H_2O\cdot THF$ dimer. Here ionization of THF initiates significant rearrangement of the dimer structure resulting in THF ring opening and finally in THF ring-break. In **b, c, d** the white, gray, and red balls represent to hydrogen, carbon, and oxygen atoms, respectively. In **d**, the green balls labeled $e_1$ and $e_2$, and the green lines indicate electrons and their trajectories, respectively.

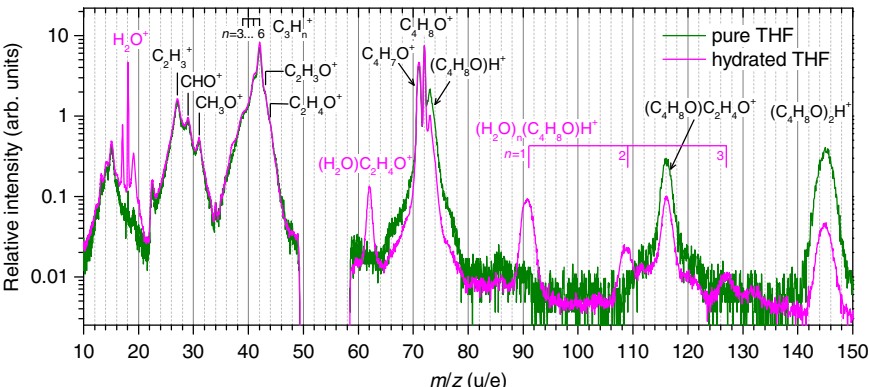

**Fig. 2 Mass-over-charge spectra.** The measurements are for pure (green line) and hydrated THF clusters (magenta line) upon electron ionization. The peaks are labeled with the assigned fragment formula. Both spectra include ion fragments from THF monomers. These are the lines in the mass region from 24 to 46 u, as well as the $C_4H_7O^+$ and $C_4H_8O^+$ ion signals. The spectra are normalized at mass 72 u corresponding to the intact $C_4H_8O^+$ cation. The experiment has no detection efficiency in the mass region from 49 to 58 u. The branching ratios of fragment species can be seen from Supplementary Note 6. Source data are provided as a Source Data file.

and subsequent dissociation of the resulting linear molecule[31]. For higher internal energies, additional hydrogen atoms can be abstracted which manifests as series of lines in the mass spectrum, e.g., $C_3H_n^+$ with $n = 3$–6.

The fragments that can be assigned to originate from clusters in the THF jet are the protonated cations $(C_4H_8O)_m\cdot H^+$ for $m = 1, 2$ and $C_4H_8O\cdot C_2H_4O^+$ from a ring-break reaction. For hydrated clusters, we observe the protonated ions $(H_2O)_n\cdot C_4H_8O\cdot H^+$ for $n = 1$–3 and also $H_2O\cdot C_2H_4O^+$, which is the equivalent ring-break reaction as the one seen in the THF dimer. These reactions will be discussed in more detail below.

It can also be seen from Fig. 2 that the fragmentation patterns in the mass region from 24 to 46 u and the $C_4H_7O^+$ ion signals

do not change significantly between the results of pure and hydrated THF clusters. This suggests that they are mainly attributed to the fragmentation processes of THF monomers.

**Identification of the ionized molecular orbitals.** In the following, we focus on the channels giving rise to the complete ring-break of THF molecule in monomers and clusters, i.e., $C_2H_4O^+$, $H_2O\cdot C_2H_4O^+$, and $C_4H_8O\cdot C_2H_4O^+$ fragments. Although this is the only ring-break reaction found for dimers, it is a rather minor channel for the monomer and superimposed by additional H-loss, namely $C_2H_3O^+$. Figure 3 presents the measured BE spectra for the three channels. Also included in the figure are the BE

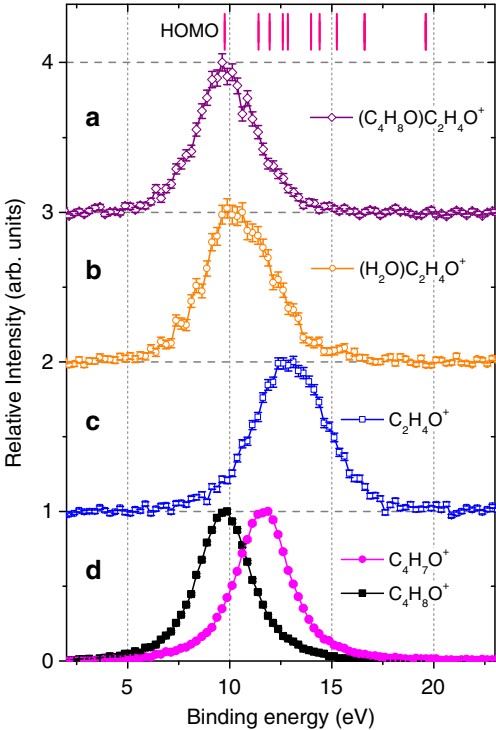

**Fig. 3 Measured binding energy spectra for various fragment species. a, b** ionization of THF·THF (**a**) and $H_2O$·THF (**b**) dimers and subsequent dissociation to $C_4H_8O$·$C_2H_4O^+$ and $H_2O$·$C_2H_4O^+$ channels, respectively. **c, d** ionization of THF monomer and subsequent dissociation to $C_2H_4O^+$ (**c**) and $C_4H_7O^+$, and non-dissociated $C_4H_8O^+$ (**d**) channels. The vertical lines on the top of the figure are the valence orbital ionization energies of the THF monomer. HOMO refers to the highest occupied molecular orbital. The statistical error bars shown correspond to the 1σ confidence interval. Source data are provided as a Source Data file.

spectra of $C_4H_7O^+$ and $C_4H_8O^+$ channels from monomers. These show peaks located at 9.8 and 11.6 eV (see Fig. 3d), respectively. The peak width of about 3.1 eV (full-width at half maximum, FWHM) is mainly attributed to the experimental BE resolution (2.9 eV, FWHM), which was determined by a measurement with He gas. The measured BE peak locations are in good agreement with previous studies of ionization and dissociation of THF monomer[31,32] showing that the HOMO ionization of THF leads to the intact $C_4H_8O^+$ ion, whereas the so-called α-cleavage reaction channel, i.e., $C_4H_7O^+$ or $(THF-H)^+$, originates mainly from ionization of the THF HOMO-1 orbital. As mentioned above the ring-break reaction in the monomer requires significantly more energy and the BE for $C_2H_4O^+$ is broader and peaking around 13 eV in Fig. 3c corresponding to the ionization of the HOMO-3 and HOMO-4 orbitals. Interestingly, the same ring-break reactions in the THF·THF and $H_2O$·THF dimers, which give rise to the $C_4H_8O$·$C_2H_4O^+$ and $H_2O$·$C_2H_4O^+$ ion fragments, show significantly lower BE with peaks located at about 9.8 and 10.4 eV, respectively (Fig. 3a, b). This indicates that these fragments are the result of HOMO ionization. Here the BE difference for the $H_2O$·$C_2H_4O^+$ channel, which is about 0.6 eV higher than the HOMO ionization energy of THF (~9.8 eV), is likely due to the modified HOMO BE in hydrated dimers. Our calculated vertical ionization energies (VIEs) are shown in Fig. 4. The VIE of the $H_2O$·THF dimer (9.95 eV) is about 0.43 eV higher than the VIE of THF monomer (9.52 eV) using the same quantum-chemistry method (Fig. 4a and b). The measured BE spectrum for $C_4H_8O$·$C_2H_4O^+$ fragment channel is nearly the same as the intact $C_4H_8O^+$ channel of THF monomer. This is in

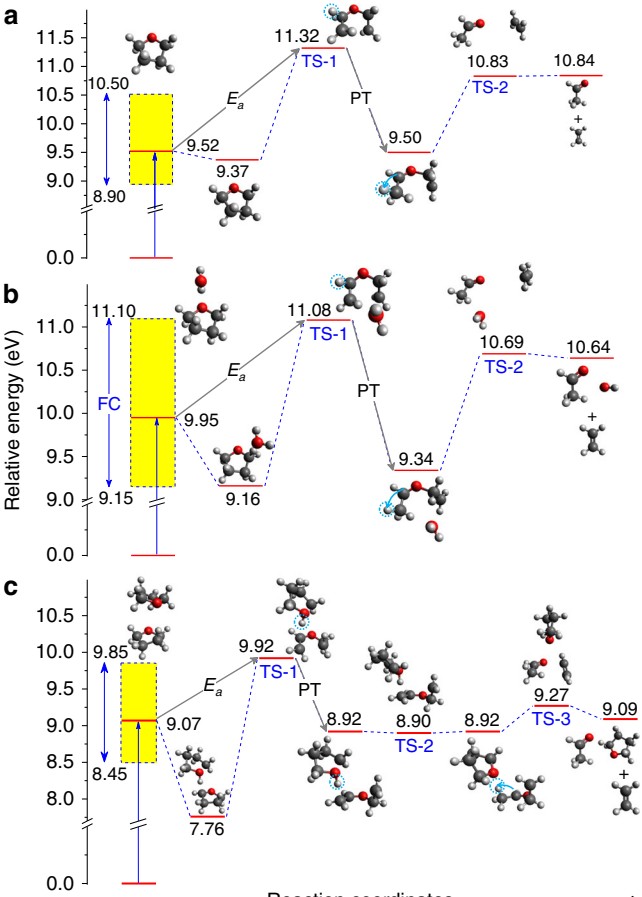

**Fig. 4 Calculated potential energy levels (eV) of ionized pure and hydrated THF systems. a–c** The energetics of $THF^+$ (**a**), $H_2O$·$THF^+$ (**b**), and THF·$THF^+$ (**c**) relative to the neutral ground state. Upon ionization of the HOMO, $H_2O$·$THF^+$ and THF·$THF^+$ can relax to the highest transition state (TS), i.e., TS-1, and then rearrange significantly the molecular structures involving particularly proton transfer (PT) and finally dissociate into $H_2O$·$C_2H_4O^+$ and $C_4H_8O$·$C_2H_4O^+$ ion fragments, respectively, and a $C_2H_4$ neutral part. For $THF^+$, the activation energy ($E_a$) from the vertical ionization point to the TS-1 state is much higher compared with the dimer systems. Thereby, the subsequent dissociation processes are not likely for the isolated $THF^+$ cation. FC refers to the Franck–Condon region, which is marked by the yellow bar. All energies include zero-point vibrational energy corrections. For comparisons between THF monomer and THF·THF dimer, the energy levels in THF monomer are recalculated using the same quantum-chemistry method as for the THF·THF dimer which is shown in Supplementary Fig. 4. The white, gray, and red balls represent hydrogen, carbon, and oxygen atoms, respectively. The PT process is shown by the circled hydrogen and arrow.

agreement with our calculation obtaining a VIE of THF·THF dimer (9.07 eV) close to the VIE of the THF monomer (9.15 eV) using the same method as for the THF·THF dimer (see Supplementary Fig. 4). Thereby, we can conclude that the two ring-break channels, i.e., $C_4H_8O$·$C_2H_4O^+$ and $H_2O$·$C_2H_4O^+$, are formed upon ionization of the HOMO in the THF site of dimers. Whereas in the monomer such ring-break channel, i.e., $C_2H_4O^+$, is associated with the ionization of HOMO-3 and HOMO-4 orbitals[32], as seen in Fig. 3c.

**Theoretical analysis of the relevant reaction pathways.** To uncover the underlying mechanism of the present observations, the TSs that are passed through in the progressing reaction after

ionization of the THF HOMO are analyzed using high-level ab-initio calculations (see Methods and Supplementary Notes 1 and 2). We calculated the potential energy surface (PES) of various intermediates to determine the reaction pathway using a relaxed PES scan. The reaction pathway is further confirmed by an intrinsic reaction coordinate calculation. The energy levels of various intermediate stationary points are illustrated in Fig. 4a–c for the three systems that are calculated using the coupled cluster single-double and perturbative triple [CCSD(T)] method with aug-cc-pVTZ basis set for the systems of THF monomer and $H_2O \cdot THF$ dimer. For the $THF \cdot THF$ dimer we had to use a more simple basis set of cc-pVDZ because of the computational complexity of this system. All energies include the zero-point vibrational energy correction and the ionization energies are obtained for the full Franck–Condon (FC) region (see Supplementary Notes 3 and 4).

The calculations show that the reaction can occur only if the highest TS corresponding to the opening of $C_\beta$–$C_\beta$ bond, which is in the ring opposite the oxygen atom, is overcome in the first step (TS-1 state, in Fig. 4a–c). Once the cation reaches this TS-1 state, there is significant rearrangement of the molecular structure during the molecular relaxation leading to the complete breakup of THF ring structure. We emphasize particularly the important role of PT in the ring-break process. After TS-1, PT is taking place from the bond position of $C_\alpha$ to $C_\beta$ of the ionized THF site (see Fig. 4). This releases some amount of internal energy to the system, which is about 1.74 eV for $H_2O \cdot THF^+$ and 1.0 eV for $THF \cdot THF^+$. The released energy accelerates the final dissociation of the cations.

It is shown in Fig. 4 that the energy levels of cations are strongly dependent on whether the THF molecule has a hydrogen-bonded neighbor, i.e., if it is embedded in a chemical environment. The calculated activation energy ($E_a$), corresponding to the required energy from vertical transition point at equilibrium geometry to the highest TS, is about 1.8 eV for the isolated $THF^+$. This energy barrier is too high for $THF^+$ to reach the TS-1 even considering the energy width of the FC region. Thus, the dissociation channel is completely closed in the THF monomer upon ionization of the HOMO. The activation energy is reduced by a factor of about two in the dimer systems, i.e., $E_a = 1.13$ eV and 0.85 eV for $H_2O \cdot THF^+$ and $THF \cdot THF^+$, respectively. Part of this reduction is due to the energy gain from a rearrangement of the dimer ion geometry following ionization. This is because the THF HOMO corresponds to the lone pair orbital of oxygen which participates in the hydrogen bonding with the neighbor and after ionization the dimer equilibrium geometry changes considerably. Therefore, the rearranging ion is in a vibrationally excited state and this internal energy facilitates reaching the TS-1. For the THF dimer after ionization there is not only rearrangement but also PT to the neighboring molecule, which contributes to the energy gain. Here, one should consider that the internal energy after ionization depends on the considered geometry of the neutral dimer. If we assume for the THF dimer geometry instead of the present stacked arrangement a T-like geometry as it was obtained by some groups[33], then a vertical transition energy of 9.71 eV is obtained which is even closer to the TS-1 energy. The FC region is shown by the yellow bands in Fig. 4. The vertical transition point at the high edge of the FC region is also comparable in energy with the TS-1 in both $H_2O \cdot THF^+$ and $THF \cdot THF^+$ systems, indicating that the ring-break processes are more likely to occur in dimers.

## Discussion
Our experimental technique as well as our calculations do not allow all possible ionization channels of the dimer systems to be examined comprehensively. Nevertheless, the current results demonstrate that embedding a molecule into a chemical environment like water can increase the biological effectiveness of ionization-induced damage to the molecule. In the present work we mainly consider small systems with only one hydrogen-bonded neighboring molecule for which high-level ab-initio calculations can be performed. By using THF as a model system of the DNA sugar-ring structure it is shown that the ring-breaking channel is activated when a water molecule is attached to THF via hydrogen bonding and an electron is removed from the HOMO of THF. This suggests that water acts as a catalyst for damaging the ring structure of THF initiated by electron ionization. Through high-level ab-initio calculations we found that the potential energy levels of various TSs are significantly modified by the water environment. This induces geometrical rearrangement and via several intermediate states finally leads to the dissociation channel. We noticed that during the molecular rearrangement PT takes place and releases internal energy to the system, which is a key point to accelerate the ring-break of the system. Similar mechanisms were found to act also in the pure THF dimer. This implies that the stability of molecular covalent bonds can be seriously affected by adding a hydrogen-bonded molecule to the system. It is also to be noted that our measurements on other hydrogen-bonded organic systems such as ethanol dimers show a similar behavior. Although HOMO ionization in the monomer produces stable cations, in the dimer C–C bond breaking in one ethanol molecule is activated (see Supplementary Note 7). This indicates that the present observation might be a general phenomenon in the ionization-induced fragmentation of hydrogen-bonded systems.

Finally, we consider a more realistic environment in the energy calculations by adding more water molecules ($nH_2O \cdot THF$, $n$ up to 4) to THF and, furthermore, we include the solvent effect using the polarizable continuum model (see Supplementary Note 5). It was found that in both cases the activation energy is reduced in comparison with THF monomer and stays almost constant for the larger hydration numbers ($n > 2$) (see Supplementary Fig. 5). This means that the catalysis effect energetically may still be open in the micro-solvated environment.

These findings at first sight contradict existing studies, which have found that solvation shells protect biomolecules from fragmentation[8,10,12–16]. However, in the systems studied here, the mechanism reducing the energy barrier to the ring opening partly is the population of vibrationally excited states of the dimer ion. If the THF molecule is embedded in a complete solvation shell, the vibrational energy can be redistributed among more degrees of freedom or even be dissipated by evaporating water molecules such that the ring-break reaction is quenched. Therefore, the protection effect may appear to compete with the catalysis effect and become dominant. The lack of larger hydrated $C_2H_4O^+$ species such as $(H_2O)_2 \cdot C_2H_4O^+$ and $(H_2O)_3 \cdot C_2H_4O^+$ at the mass-over-charge values of 80 and 98, respectively, in Fig. 2 is a signature of such protection effect in our experiment. Thus, the occurrence of the THF ring-break reaction strongly depends on the spatial geometry and mobility of the molecules in the local environment. These results could have implications for our understanding of ionization damage in biological matter.

The hydrogen bonding is an important noncovalent interaction ubiquitous in nature from base-pair interactions in DNAs and sophisticated supramolecular assemblies to the dense and cold molecular clouds in outer space and planetary atmospheres[34–36]. The present electron-collision induced reactions in molecules are common phenomena in many fields of science and technology, in the gas phase and in the condensed phase or at interfaces. Our studies concerning $THF \cdot THF$ and $H_2O \cdot THF$ dimers clearly demonstrate that hydrogen-bonded molecules can considerably affect the neighboring molecule and play the role of a catalyst for the break-up of its covalent bonds.

## Methods

**Coincident ion and electron detection**. The experimental data were obtained by crossing an electron projectile beam with a gas target jet and employing a multi-particle coincidence spectrometer (reaction microscope) for detection of the charged collision fragments[29,30]. The pulsed electron beam is generated in an electron gun from a tantalum photocathode, which is irradiated by UV-light pulses of 0.5 ns duration and it is collimated by electrostatic lens elements. The energy width of the electron beam is about 0.5 eV[31]. It is guided by an axial magnetic field (0.7 mT) to the crossing zone with the gas jet and further to a beam dump which is a central bore in the electron detector. Secondary electrons and ions are extracted by means of a homogeneous electric field to opposite directions and projected onto two position- and time-sensitive multi-channel plate detectors with 80 mm diameter of the active area. From the impact positions and the times-of-flight, the momentum vectors and consequently the kinetic energies of the particles emerging from the reaction are determined. The acceptance angle for detection of low-energy electrons up to the kinetic energy of 15 eV is almost $4\pi$ with exception of small forward and backward angles, which are lost due to the detector bore. Projectile electrons that have undergone ionizing collisions are detected for scattering angles up to about 35°. To maximize the acceptance for molecular ion fragments, the electric extraction field of 1.0 V cm$^{-1}$ is ramped up to 20 V cm$^{-1}$ after 400 ns when the electrons have reached the detector. In our experiment, monomer ionization is simultaneously recorded with cluster ionization.

**Cluster production**. The pure and hydrated THF clusters are generated in a supersonic expansion of helium gas (stagnation pressure 1 bar), which is seeded with pure THF vapor or mixed water and THF vapor. To pick up the target molecules, the helium gas is guided through one or two reservoirs containing liquid THF and water. For production of pure THF clusters, only the THF reservoir is used. The hydrated clusters are created with the two reservoirs containing water and THF at temperatures of 60 and 25 °C (room temperature), respectively. The gas mixture is expanded through a 30 μm nozzle, which is heated to 100 °C into the vacuum. The gas beam is collimated by two skimmers with 200 μm diameter aperture at their apex, and located ~3 mm and 20 mm downstream from the nozzle. The relative fraction of pure and mixed THF dimers in the jet as given in the text was estimated from the relative intensities of the respective ion species produced by HOMO ionization. For the THF monomer, this is $C_4H_8O^+$. The related line in the TOF spectrum in Fig. 2 is comparatively narrow showing that this ion is not resulting from dissociation of clusters. For the THF$_2$ dimer, the ion species produced by HOMO ionization is $C_4H_8O \cdot C_2H_4O^+$ and for the H$_2$O·THF dimer this is $H_2O \cdot C_2H_4O^+$.

**Quantum-chemistry calculations**. The calculations were carried out using the Gaussian package[37]. The ground-state equilibrium geometries of the isolated THF molecule and the hydrogen-bonding H$_2$O·THF dimer were optimized with the second-order Møller–Plesset method using the aug-cc-pVTZ basis set. For the THF·THF dimer, due to the computational complexity of this system, the Becke's three-parameter hybrid functional combined with Lee–Yang–Parr correlation functional (B3LYP) method was used together with a cc-pVDZ basis set. The neutral and singly charged electronic energies were determined using a CCSD(T) method with a aug-cc-pVTZ basis set for the systems of THF monomer and H$_2$O·THF dimer and a cc-pVDZ basis set for THF·THF dimer. The TSs and zero-point energy corrections were determined by the B3LYP method using the aug-cc-pVTZ basis set for THF, H$_2$O·THF and the cc-pVDZ basis set for THF·THF. The VIE in the FC region were calculated by outer-valence Green's function method considering the zero-point vibration by quantum harmonic oscillator distribution.

## Data availability

The source data underlying Fig. 2 and Fig. 3a–d are provided as a Source Data file. The data supporting this study are also available from the corresponding author upon reasonable request.

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

## Acknowledgements

This work was supported by the National Natural Science Foundation of China under Grants Numbers 11774281 and 11974272, and by the Deutsche Forschungsgemeinschaft (DFG) under project number RE 2966/3-1. E.W. acknowledges a fellowship from the Alexander von Humboldt Foundation.

## Author contributions

X.R. and A.D. conceived and designed the experiments. X.R. performed the experiments and analyzed the data. E.W. carried out the quantum-chemistry calculations. X.R. and E.W. wrote the first draft of the manuscript. W.B., H.R., and T.P. contributed to the interpretation of the data and edited the manuscript. All authors edited and approved the final version of the manuscript.

## Competing interests

The authors declare no competing interests.
