## [Peer Review File · Nature Communications]

Reviewers' comments:

Reviewer #1 (Remarks to the Author):

The manuscript by Wang and coworkers presents a study combining carefully obtained experimental data on electron ionization of small tetrahydrofuran clusters with theory. The topic of radiation damage to biomolecules is recent, the experimental data are of high quality, taken using state of the art technique. The observation of water mediated molecular breakup move the studies of low-energy electron induced processes in biomolecules towards new direction. Mobile proton hypothesis was so far explored for proteins and the present study transfer it also to the field of DNA damage. I evaluate the present work as very important and I highly recommend it for publication in the Nature Communications.

I have following comments to authors that may be considered before re-submitting the text.

1. In introduction, the importance of the studied process for biology is a bit exaggerated. e.g. "Damage to cells genomic DNA..... represents, certainly, a central mechanism in the pathogenesis...."

I suggest to replace "certainly" by "an important"

The complexity of the cell death or biological processes in the tissue cannot be reduced to DNA damage.

e.g. "damage to deoxyribose is known as a key intermediate...."

I suggest to replace by "may be involved"

It is mixing apples and pears when comparing OH radical reactions with deoxyribose and electron reactions with THF. Much better model for DNA damage will be hydroxyl substituents of THF or furfuryl alcohol. The mechanism, which you report, is certainly of high importance for the radiation damage but not because of THF. THF is a technical molecule, which was actually designed to easily open the ring for polymerization.

2. Figure 4. Full width figure will exclude overlaps of the fragments from one energetic level to another. Do the relaxed structure of dimer cation have opened ring structure? Is it possible that proton transfer from one molecular unit to another one promotes new fragmentation channels (e.g. reported for amino acid clusters Pouilly Chem. Phys. Chem. 16(11),2389, 2015)? Maybe you can compare your observed fragmentation pattern to MS of isolated THF.

3. p. 12 discussion second paragraph. I suggest to modify this paragraph.

First, I am sure that you see less fragmentation than in the case of isolated molecule. You can include ratio of fragment ion yield to parent ion yield for isolated and clustered THF.

Second, the paragraph concludes that the process will depend on the cluster size and maybe do not occur in bulk. Then it may be irrelevant for radiation damage. I strongly suggest you to recalculate the ionization and transition state energies in bulk water environment, which will significantly strengthen the conclusions.

4. Did you observe any fragmentation channels opened by primary ionization of water?

Reviewer #2 (Remarks to the Author):

I am not surprised that a micro-water environment (i.e., a very few water molecules) could induce crucial changes in molecular decomposition pathways when ions are involved. Thus, my lack of surprise also extends to models of the sugar-ring structure of DNA. That, I suppose, was the authors' original hypothesis. Their experimental and computational work is solid and supportive of their conclusions. Basically, I believe their conclusions. The question from the journal's perspective, however, is whether this is important enough to publish. In my opinion, I think it is, because this is news to much of the scientific community and its publication in Nature Communications may help change the way many people look at such problems. I recommend that this article be published essentially as it is.

Reviewer #3 (Remarks to the Author):

This paper reports a sophisticated experiment in which two dimers, namely tetrahydrofuran-H₂O and (tetrahydrofuran)₂, are bombarded with electrons, and the produced electrons (including the impinging one) and resulting ionic molecular fragments are detected in coincidence. The most interesting result is that the presence of a partner H₂O or THF molecule favors fragmentation in comparison with the case of a single tetrahydrofuran (THF) molecule. To understand this result, standard quantum chemistry calculations have also been performed. These allow to identify the different minima and transition states involved in the fragmentation process, to describe the corresponding energetics, and hence to propose a sound mechanism that explains the observed fragmentation. This is in itself a nice piece of work that will certainly be of interest to the chemical physics or physical chemistry communities, but not much beyond. The reasons are the following:

1/ The authors claim that "This result demonstrates that THF can be more easily broken in a chemical environment, which might offer an alternative way for highly targeted radiation therapies" (abstract), "in order to mimic a biological environment" (introduction), "These observations reveal a so-far unnoticed role of the water environment in enhancing the complete ring-break of the THF molecule" (page 5), and "This suggests that water acts as a catalyst for damaging the ring structure of THF" (conclusion, page 12). If such claims were unambiguously supported by the results presented in the manuscript, I would not have any doubt to recommend publication in Nature Comm. However, they are not supported. Indeed, adding a SINGLE water molecule to THF does not mimic at all the effect of a water environment. First, for obvious chemical reasons, the added water molecule does not have any choice but to bind THF in the exact position where THF will break. In the presence of many water molecules (i.e., a real water environment), this binding may occur, but it will be strongly affected by the other surrounding moving molecules. Second, if a similar electron collision experiment was performed on a THF molecule surrounded by many water molecules, the most likely process would be to ionize water, not THF (it's pure statistics). The produced electron could still attack THF, but this would not be the most likely process, since it is well known that ionization of water leads to the formation of OH radicals that are extremely reactive and can therefore induce a completely different chemistry (the literature on this topic is abundant, both covering experimental and theoretical work, to be explicitly quoted here). Furthermore, electrons ejected from a water molecule will mostly collide with other water molecules, because they are much more abundant, and this should be considered. Therefore, all the claims suggesting that their results can be extrapolated to a real water environment are not sustained. Most likely, they are only valid for the particular dimer they have considered.

2/ The authors claim in several parts of the manuscript that they perform "ab-initio dynamical calculations" (abstract), "molecular dynamics simulations" (main text, page 5), "ab-initio dynamical calculations" (discussion, page 12). If this was true, such calculations would have been a real added value to the manuscript. However, all theoretical results reported in the manuscript come from standard STATIONARY (not dynamical) quantum chemistry calculations, such as MP2, CCSD(T), DFT-B3LYP, OVGf, etc, which have been most likely performed by using a standard computational package for molecular structure calculations (I guess it could be Gaussian, but no reference to the computational package used for these calculations is given in the manuscript; only Newton-X is referred to in the SM for a very specific type of calculation). Nowhere in the manuscript I have seen results of calculations in which the variable "time" has been taken into account, as e.g. in methods such as time-dependent DFT, Born-Oppenheimer molecular dynamics (BOMD), Atom Centered Density Matrix Propagation molecular dynamics (ADMP), trajectory surface hopping (TSH), or the like. Therefore, it is not fair to say that the present manuscript reports the results of "dynamical" calculations.

In summary, this work reports the results of sophisticated experiments for THF dimers containing a single water molecule, supported by standard electronic structure quantum chemistry calculations, which are interesting in themselves, but do not provide information on the role of a real water environment and do not provide either the promised "dynamical" explanation (since all

calculations were performed by using stationary methods). For these reasons, I recommend the authors to focus the presentation and ulterior analysis and conclusions in what they have really done, and submit the manuscript to a chemical physics or a physical chemistry journal.

Response to Referees

Reviewer #1 (Remarks to the Author):

The manuscript by Wang and coworkers presents a study combining carefully obtained experimental data on electron ionization of small tetrahydrofuran clusters with theory. The topic of radiation damage to biomolecules is recent, the experimental data are of high quality, taken using state of the art technique. The observation of water mediated molecular breakup move the studies of low-energy electron induced processes in biomolecules towards new direction. Mobile proton hypothesis was so far explored for proteins and the present study transfer it also to the field of DNA damage.

Comment 1: I evaluate the present work as very important and I highly recommend it for publication in the Nature Communications.

Our reply: *We thank the Reviewer for the very positive evaluation of our manuscript.*

Comment 2: I have following comments to authors that may be considered before re-submitting the text.

a) In introduction, the importance of the studied process for biology is a bit exaggerated. e.g. "Damage to cells genomic DNA..... represents, certainly, a central mechanism in the pathogenesis...."

b) I suggest to replace "certainly" by "an important"

c) The complexity of the cell death or biological processes in the tissue cannot be reduced to DNA damage.

d) e.g. "damage to deoxyribose is known as a key intermediate.....". I suggest to replace by "may be involved". It is mixing apples and pears when comparing OH radical reactions with deoxyribose and electron reactions with THF. Much better model for DNA damage will be hydroxyl substituents of THF or furfuryl alcohol. The mechanism, which you report, is certainly of high importance for the radiation damage but not because of THF. THF is a technical molecule, which was actually designed to easily open the ring for polymerization.

Our reply: *In the present manuscript, we revised the introduction section according to the referee's comments and suggestions. We weakened the close connection to radiotherapy. The sentences related to a) - d) have been omitted. We now focus on the fundamental mechanism of molecular break-up as well as the newly observed catalytic effect of the chemical environment, i.e. the breaking of molecular covalent bond is activated by the hydrogen bonding environment upon ionization of the outermost molecular orbital (HOMO). (Paragraphs 1-2 in Introduction section). In addition, we performed new experiments of other organic system. Measurements on the ionization and fragmentation of ethanol monomer and dimers reveal the similar effect as our studies on THF, i.e. the C-C bond breaking is activated in ethanol dimer (CH₃CH₂OH)₂ upon HOMO ionization. This proves that the present observation might be a general phenomenon in the ionization-induced fragmentation of hydrogen bonding systems. (Lines 228-*

232 in Discussion section, the experimental data are presented in Fig. S6 of Supplementary Information)

Comment 3: **a)** Figure 4. Full width figure will exclude overlaps of the fragments from one energetic level to another. **b)** Do the relaxed structure of dimer cation have opened ring structure? **c)** Is it possible that proton transfer from one molecular unit to another one promotes new fragmentation channels (e.g. reported for amino acid clusters Pouilly Chem. Phys. Chem. 16(11),2389, 2015)? **d)** Maybe you can compare your observed fragmentation pattern to MS of isolated THF.

Our reply:

- a) We modify the size of Fig. 4 to avoid the overlaps.
- b) The geometry relaxation of the dimer cation does not have opened ring structure.
- c) For the intermolecular proton transfer induced new fragmentation channels, we can confirm that: (I) very strong intensity of protonated species are observed in our mass spectrum, i.e. $(C_4H_8O)H^+$, $(H_2O)_n(C_4H_8O)H^+$, and $(C_4H_8O)_2H^+$, etc. (II) In our previous work, one fragment was presented at $m/z=55$ (X. Ren, et al., J. Chem. Phys. 141, 134314 (2014)), and this ionic product is assigned to the fragmentation of ionized THF clusters by a recent experiment work (M. Neustetter, et al, J. Am. Soc. Mass Spectrom. 28, 866 (2017)). Although the binding energy spectrum for this ion can be obtained from our experiment, the detailed mechanism of this ionic fragment is not clear so far.
- d) We include the branching ratios of fragments for THF^+ in the Supplementary Information. We can compare the fragmentation pattern of hydrated THF and pure THF clusters to that of isolated THF in the mass range from 24 to 50. The intact $C_4H_8O^+$ ion yields are normalized to 1, the total fragment ion yields from 24 to 50 are about 6.82 for isolated THF (M. Dampe, et al., J. Phys. B: At. Mol. Opt. Phys. 44 (2011) 055206), 6.98 for pure THF clusters, and 7.43 for hydrated THF. This means that the fragmentation patterns in the mass range from 24 to 50 can be most likely due to the dissociation of isolated THF. While slightly increased intensity in hydrated THF can be partly ascribed to the water clusters ion fragments like $(H_2O)_2H^+$ at m/z 37.

Comment 4: p. 12 discussion second paragraph. I suggest to modify this paragraph.

a) First, I am sure that you see less fragmentation than in the case of isolated molecule. You can include ratio of fragment ion yield to parent ion yield for isolated and clustered THF.

b) Second, the paragraph concludes that the process will depend on the cluster size and maybe do not occur in bulk. Then it may be irrelevant for radiation damage. I strongly suggest you to recalculate the ionization and transition state energies in bulk water environment, which will significantly strengthen the conclusions.

Our reply:

- a) The branching ratios of fragments to THF^+ are shown in the Supplementary Information (Table S21).
- b) We added new theoretical results, firstly, for larger clusters ($nH_2O \cdot THF^+$, n up to 4) and,

secondly, including additionally the solvation effect by a polarizable continuum model (PCM). We find that in both models the activation energies are reduced compared to the THF monomer. If the hydration number is larger than 2, the activation energy is almost a constant for every model. Thereby the new calculations support our conclusion that the water environment can show a catalytic effect on the ring opening of THF cation. However, as we discussed in the third paragraph of the DISCUSSION section, if the THF molecule is embedded in a complete solvation shell, the protection effect may be dominant since the internal vibrational energy can be redistributed among more degrees of freedom or even dissipated by evaporation of water molecules in case of a cluster.

Comment 5: Did you observe any fragmentation channels opened by primary ionization of water?

Our reply: *If the water molecule is ionized, there can be proton transfer from water cation to THF. As seen in the mass spectra, there are the protonated species which can be contributed by primary ionization of water, e.g. $(C_4H_8O)H^+$, $(H_2O)_n(C_4H_8O)H^+$.*

Reviewer #2 (Remarks to the Author):

Comment: I am not surprised that a micro-water environment (i.e., a very few water molecules) could induce crucial changes in molecular decomposition pathways when ions are involved. Thus, my lack of surprise also extends to models of the sugar-ring structure of DNA. That, I suppose, was the authors' original hypothesis. Their experimental and computational work is solid and supportive of their conclusions. Basically, I believe their conclusions. The question from the journal's perspective, however, is whether this is important enough to publish. In my opinion, I think it is, because this is news to much of the scientific community and its publication in Nature Communications may help change the way many people look at such problems. I recommend that this article be published essentially as it is.

Our reply: *We thank the Reviewer for the very positive evaluation of our manuscript.*

Reviewer #3 (Remarks to the Author):

This paper reports a sophisticated experiment in which two dimers, namely tetrahydrofuran-H₂O and (tetrahydrofuran)₂, are bombarded with electrons, and the produced electrons (including the impinging one) and resulting ionic molecular fragments are detected in coincidence. The most interesting result is that the presence of a partner H₂O or THF molecule favors fragmentation in comparison with the case of a single tetrahydrofuran (THF) molecule. To understand this result, standard quantum chemistry calculations have also been performed. These allow to identify the different minima and transition states involved in the fragmentation process, to describe

the corresponding energetics, and hence to propose a sound mechanism that explains the observed fragmentation. This is in itself a nice piece of work that will certainly be of interest to the chemical physics or physical chemistry communities, but not much beyond. The reasons are the following:

Comment 1: 1/ The authors claim that “This result demonstrates that THF can be more easily broken in a chemical environment, which might offer an alternative way for highly targeted radiation therapies” (abstract), “in order to mimic a biological environment” (introduction),

Our reply: *The Abstract section has been modified accordingly. We now focus more on the fundamental mechanism of molecular break-up as well as the newly observed catalytic effect of the chemical environment. The last sentences of the abstract now read: “These results demonstrate that the local environment in form of a hydrogen bonded water or THF molecule can considerably affect the stability of molecular covalent bonds.” and the sentence in the Introduction “in order to mimic a chemical environment.” (line 69).*

Comment 2: “These observations reveal a so-far unnoticed role of the water environment in enhancing the complete ring-break of the THF molecule” (page 5), and “This suggests that water acts as a catalyst for damaging the ring structure of THF” (conclusion, page 12). If such claims were unambiguously supported by the results presented in the manuscript, I would not have any doubt to recommend publication in Nature Comm. However, they are not supported.

Our reply: *This comment most likely refers to **Comment 3**. The referee considers one water molecule bonded to THF not as a water or aqueous environment. As we elucidate in the reply to **Comment 3** in the revised manuscript we have added theoretical results for clusters with more water molecules and bulk effects. And also we now focus more on the fundamental, new molecular breaking mechanism, i.e. the breaking of molecular covalent bond is activated by the hydrogen bonding environment upon ionization of the outermost molecular orbital (HOMO). (Paragraphs 1-2 in Introduction section & the whole Discussion section). In addition, we performed new experiments of other organic system. Measurements on the ionization and fragmentation of ethanol monomer and dimers reveal the similar effect as our studies on THF, i.e. the C-C bond breaking is activated in ethanol dimer (CH₃CH₂OH)₂ upon HOMO ionization. This proves that the present observation might be a general phenomenon in the ionization-induced fragmentation of hydrogen bonding systems. (Lines 228-232 in Discussion section, the experimental data are presented in Fig. S6 of Supplementary Information)*

Comment 3: Indeed, adding a SINGLE water molecule to THF does not mimic at all the effect of a water environment. First, for obvious chemical reasons, the added water molecule does not have any choice but to bind THF in the exact position where THF will break. In the presence of many water molecules (i.e., a real water environment), this binding may occur, but it will be strongly affected by the other surrounding moving molecules.

Our reply: We agree that adding a SINGLE water molecule to THF does not mimic the full effect of a water environment. Nevertheless, it is an important step to go beyond the isolated system. In the present work, we aim to study the hydrogen bonding effects in a small system. The reasons are two folds: 1) For studying the ionization process experimentally, a kinematically complete measurement of both electrons and the ion is essential in order to obtain the most detailed information about the reaction mechanism. Such an experiment is feasible for small systems in gas phase but not for a bulk water environment. 2) The higher level ab initio calculations can be performed in such smaller system for a better understanding of the experimental observations.

As a consequence of the criticism, in the revised manuscript, we consider a more realistic environment in the energy calculations by adding more water molecules ($n\text{H}_2\text{O}\cdot\text{THF}$, n up to 4) to THF and include the solvent effect using the polarizable continuum model (PCM). It is found that in both cases the activation energy is reduced in comparison with THF monomer and stays almost constant for the larger hydration number ($n>2$) (see Fig. S5 in the Supplementary Information). This means that the catalytic effect may still exist in the micro-solvated environment.

On the other hand, in larger clusters the protection effect of the water environment may compete with the present catalytic effect and quench the reaction. The internal energy of the cluster cation can be redistributed among more degrees of freedom or even be dissipated by evaporating water molecules. The lack of larger hydrated $\text{C}_2\text{H}_4\text{O}^+$ species like $(\text{H}_2\text{O})_2\cdot\text{C}_2\text{H}_4\text{O}^+$ and $(\text{H}_2\text{O})_3\cdot\text{C}_2\text{H}_4\text{O}^+$ at the m/z of 80 and 98, respectively could be one signature of such protection effect observed in our experiment. Both, catalysis and protection effects of water environment can exist in clusters. The present catalytic effect is more important in smaller clusters which is the new observation of our study. All these aspects are discussed in the revised manuscript in the section "Discussion". (lines: 232-238, 245-252)

Comment 4: Second, if a similar electron collision experiment was performed on a THF molecule surrounded by many water molecules, the most likely process would be to ionize water, not THF (it's pure statistics). The produced electron could still attack THF, but this would not be the most likely process, since it is well known that ionization of water leads to the formation of OH radicals that are extremely reactive and can therefore induce a completely different chemistry (the literature on this topic is abundant, both covering experimental and theoretical work, to be explicitly quoted here). Furthermore, electrons ejected from a water molecule will mostly collide with other water molecules, because they are much more abundant, and this should be considered. Therefore, all the claims suggesting that their results can be extrapolated to a real water environment are not sustained. Most likely, they are only valid for the particular dimer they have considered.

Our reply: We agree that ionization of the water environment plays a different and important role in DNA damage via the formation of OH radicals. It is known that high energy radiation induced damage to DNA is caused by roughly two-thirds via OH radicals from water but still by one-third directly by energy deposition in DNA and its hydrogen bonded water (see, e.g., B. D. Michael, P. O'Neill, Science, Vol. 287, Issue 5458, p. 1603). In this work, we focus on the

damage mechanism of a small building block of a bio-molecule induced by low-energy electrons in a local environment. To compare the contributions of direct ionization of the water environment, we can consider the absolute cross sections. For the impact energy of about 65 eV, we find in literatures: (1) the total ionization cross section of water is $\sim 1.9 \times 10^{-16} \text{ cm}^2$ (Itikawa, et al, J. Phys. Chem. Ref. Data, 34, 1, (2005)); (2) the total ionization cross section of THF is $\sim 14 \times 10^{-16} \text{ cm}^2$ (Wolff, et al, J. Chem. Phys. 151, 064304 (2019)). Thereby, due to the much larger cross-sections of (2), ionization-induced damage of the bio-molecule like THF, is important in the hydrated system. The cross sections for real macro-molecules like DNA will be even orders of magnitudes larger.

Comment 5: 2/ The authors claim in several parts of the manuscript that they perform “ab-initio dynamical calculations” (abstract), “molecular dynamics simulations” (main text, page 5), “ab-initio dynamical calculations” (discussion, page 12). If this was true, such calculations would have been a real added value to the manuscript. However, all theoretical results reported in the manuscript come from standard STATIONARY (not dynamical) quantum chemistry calculations, such as MP2, CCSD(T), DFT-B3LYP, OVGF, etc, which have been most likely performed by using a standard computational package for molecular structure calculations (I guess it could be Gaussian, but no reference to the computational package used for these calculations is given in the manuscript; only Newton-X is referred to in the SM for a very specific type of calculation). Nowhere in the manuscript I have seen results of calculations in which the variable “time” has been taken into account, as e.g. in methods such as time-dependent DFT, Born-Oppenheimer molecular dynamics (BOMD), Atom Centered Density Matrix Propagation molecular dynamics (ADMP), trajectory surface hopping (TSH), or the like. Therefore, it is not fair to say that the present manuscript reports the results of “dynamical” calculations.

Our reply: *This is correct. In this work, we perform static energy calculations using the Gaussian16 package. The revised manuscript has been changed accordingly. It is noticed that the Atom Centered Density Matrix Propagation (ADMP) method has also been applied to the present systems for molecule dynamical simulations. However, the life time of the cluster cation is very long (it can be longer than 100 ps or in the ns scale). It is not possible for us to perform such dynamical calculation. This can be a future subject for professional theoreticians.*

Comment 6: In summary, this work reports the results of sophisticated experiments for THF dimers containing a single water molecule, supported by standard electronic structure quantum chemistry calculations, which are interesting in themselves, but do not provide information on the role of a real water environment and do not provide either the promised “dynamical” explanation (since all calculations were performed by using stationary methods). For these reasons, I recommend the authors to focus the presentation and ulterior analysis and conclusions in what they have really done, and submit the manuscript to a chemical physics or a physical chemistry journal.

Our reply: *In the present work, we focus on the ionization and fragmentation of a simple*

hydrogen bonding system (water mixed with a bio-molecule) induced by low-energy electrons in order to figure out the role of water environment. Fragment ion and electron coincident momentum spectroscopy, accompanied by ab-initio calculations, allow us to directly identify the ionization mechanisms. We found that the local environment introduced by a weak hydrogen bonding can considerably affect the stability of molecular covalent bonds, which is a previously unseen (new) mechanism (i.e. the catalytic effect of the water environment) for the breakup of molecular covalent bonds. In addition, Measurements on the ionization and fragmentation of ethanol monomer and dimers reveal the similar effect as our studies on THF. This proves that the present catalytic effect might be a general phenomenon in the ionization-induced fragmentation of hydrogen bonding systems. (Lines 228-232 in Discussion section) The present observation can be relevant to radiation damage of bio-molecule as well as the chemistry in the interstellar medium (astrochemistry). As small systems in a local environment are widely existing in the outer space particularly in dense and cold molecular clouds, and also at the higher density and higher temperature found in planetary atmospheres. The catalytic effect of the environment may play an important role in breaking and making chemical bond of molecules. (lines 262-266)

Thereby, we are confident that the present results will be of interest to the broad readers of Nature Communication.

Reviewer #1 (Remarks to the Author):

As already stated in my original review, the work is of high quality and will have a significant impact on the understanding of solvent effects in the reaction dynamics. Authors fully answered my questions and incorporated my comments. Therefore, I highly recommend the manuscript for publication in the Nature Communications.

Two minor comments, which still may be considered:

1. uniform the terms ring opening and breaking within text. Cleavage of one bond should be ring opening. The process in which two fragments are formed can be breaking, fragmentation or cleavage but please chose 1 term. E.g. line 79 you use cleavage, SI ring opening, further in text breaking....

2. Supplementary material Fig S6 caption does not correspond to figures and text in panels. b and c states THF monomer but in figure you have drawings and text for Ethanol.

In my opinion, ethanol data was not necessary to add. Actually, for me it is just confusing. I don't see any systematics in choosing these two molecules. Maybe as a stepwise process of ring opening and ethanol formation from sugars. But you do not see ethanol cation from THF. I think, hydroxyl-substituted furans will be much better model as I suggested in the previous report.

Reviewer #3 (Remarks to the Author):

The authors have substantially revised their manuscript following my recommendations. The generality of their findings in the context of biological applications has been conveniently downgraded and is now restricted to the case of simple hydrogen bonding systems. In turn, they show that a similar behavior is observed for ethanol monomers and dimers, which is interesting. The new version also avoids mentioning that the manuscript presents the results of dynamical theoretical calculations (which was not true) and reports additional (standard) calculations for THF molecules surrounded by up to four water molecules. So, technically speaking, the manuscript fulfills all the requirements to be published in its present form. Still, in line with the report of reviewer 2 and the narrower focus of the present version, I am not sure that Nature Communications is the most appropriate journal to publish this work.

Response to Referees

Reviewer #1 (Remarks to the Author):

As already stated in my original review, the work is of high quality and will have a significant impact on the understanding of solvent effects in the reaction dynamics. Authors fully answered my questions and incorporated my comments. Therefore, I highly recommend the manuscript for publication in the Nature Communications.

Our reply:

We thank the Reviewer for the very positive evaluation of our manuscript.

Two minor comments, which still may be considered:

1. uniform the terms ring opening and breaking within text. Cleavage of one bond should be ring opening. The process in which two fragments are formed can be breaking, fragmentation or cleavage but please chose 1 term. E.g. line 79 you use cleavage, SI ring opening, further in text breaking....

Our reply:

We uniformed the terms according to the referee's recommendation in the manuscript. For cleavage of one bond we use ring-opening and for the process where two fragments are formed we use ring-break.

2. Supplementary material Fig S6 caption does not correspond to figures and text in panels. b and c states THF monomer but in figure you have drawings and text for Ethanol.

In my opinion, ethanol data was not necessary to add. Actually, for me it is just confusing. I don't see any systematics in choosing these two molecules. Maybe as a stepwise process of ring opening and ethanol formation from sugars. But you do not see ethanol cation from THF. I think, hydroxyl-substituted furans will be much better model as I suggested in the previous report.

Our reply:

- We revised the figure caption accordingly.*
- We thank the referee for the suggestions. We agree that THF and ethanol are two different organic molecules. In the Supplementary Information, we refer to the ethanol dimers as another example of hydrogen-bonded system. A similar behavior like in THF dimers was observed, i.e. in the dimer the neighbouring molecule plays the role as a catalyst for the break-up of its covalent bonds (C-C bond in ethanol). This might be interesting to the general reader. In order to make this motivation clearer we modified the respective text in the manuscript (text starting page 12, line 223).*

Reviewer #3 (Remarks to the Author):

The authors have substantially revised their manuscript following my recommendations. The generality of their findings in the context of biological applications has been conveniently downgraded and is now restricted to the case of simple hydrogen bonding systems. In turn, they show that a similar behavior is observed for ethanol monomers and dimers, which is interesting. The new version also avoids mentioning that the manuscript presents the results of dynamical theoretical calculations (which was not true) and reports additional (standard) calculations for THF molecules surrounded by up to four water molecules. So, technically speaking, the manuscript fulfills all the requirements to be published in its present form. Still, in line with the report of reviewer 2 and the narrower focus of the present version, I am not sure that Nature Communications is the most appropriate journal to publish this work.

Our reply:

We thank the Reviewer for the positive evaluation of our manuscript.